# Prognostic Significance of the Relative Load of KPC-Producing *Klebsiella pneumoniae* within the Intestinal Microbiota in a Prospective Cohort of Colonized Patients

Elena Pérez-Nadales,[a,b,c] Alejandra M. Natera,[a,b] Manuel Recio-Rufián,[a,b,d] Julia Guzmán-Puche,[a,b,d] Juan Antonio Marín-Sanz,[b] Carlos Martín-Pérez,[e] Ángela Cano,[a,b,d] Juan José Castón,[a,b,c,d] Cristina Elías-López,[a,b] Isabel Machuca,[a,b,d] Belén Gutiérrez-Gutiérrez,[a,f] Luis Martínez-Martínez,[a,b,c,d] Julián Torre-Cisneros[a,b,c,d]

[a]Spanish Network for Research in Infectious Diseases (REIPI), Centro de Investigación Biomédica en Red de Enfermedades Infecciosas (CIBERINFEC), Instituto de Salud Carlos III, Madrid, Spain

[b]Infectious Diseases (GC-03) and Clinical and Molecular Microbiology (GC-24) Groups, Maimonides Biomedical Research Institute of Cordoba, Reina Sofía University Hospital, University of Cordoba (IMIBIC/HURS/UCO), Cordoba, Spain

[c]Department of Agricultural Chemistry, Edaphology and Microbiology and Department of Medical and Surgical Sciences, University of Cordoba, Cordoba, Spain

[d]Clinical Units of Infectious Diseases and Microbiology, Reina Sofía University Hospital, Cordoba, Spain

[e]Doctor in Medicine, specialist in Family and Community Medicine in the Andalusian Health Service, Granada, Spain

[f]Clinical Unit of Infectious Diseases, Microbiology and Preventive Medicine, University Hospital Virgen Macarena, Institute of Biomedicine of Seville (IBiS), Seville, Spain

Elena Pérez-Nadales and Alejandra M. Natera contributed equally to this article. Author order was determined in order of decreasing seniority.

**ABSTRACT**   Increased relative bacterial load of KPC-producing *Klebsiella pneumoniae* (KPC-KP) within the intestinal microbiota has been associated with KPC-KP bacteremia. Prospective observational study of KPC-KP adult carriers with a hospital admission at recruitment or within the three prior months (January 2018 to February 2019). A qPCR-based assay was developed to measure the relative load of KPC-KP in rectal swabs ($RL_{KPC}$, proportion of $bla_{KPC}$ relative to 16S rRNA gene copy number). We generated Fine-Gray competing risk and Cox regression models for survival analysis of all-site KPC-KP infection and all-cause mortality, respectively, at 90 and 30 days. The median $RL_{KPC}$ at baseline among 80 KPC-KP adult carriers was 0.28% (range 0.001% to 2.70%). Giannella Risk Score (GRS) was independently associated with 90-day and 30-day all-site infection (adjusted subdistribution hazard ratio [aHR] 1.23, 95% CI = 1.15 to 1.32, $P < 0.001$). $RL_{KPC}$ (adjusted hazard ratio [aHR] 1.04, 95% CI = 1.01 to 1.07, $P = 0.008$) and age (aHR 1.05, 95% CI = 1.01 to 1.10, $P = 0.008$) were independent predictors of 90-day all-cause mortality in a Cox model stratified by length of hospital stay (LOHS) ≥20 days. An adjusted Cox model for 30-day all-cause mortality, stratified by LOHS ≥14 days, included $RL_{KPC}$ (aHR 1.03, 95% CI = 1.00 to 1.06, $P = 0.027$), age (aHR 1.10, 95% CI = 1.03 to 1.18, $P = 0.004$), and severe KPC-KP infection (INCREMENT-CPE score >7, aHR 2.96, 95% CI = 0.97 to 9.07, $P = 0.057$). KPC-KP relative intestinal load was independently associated with all-cause mortality in our clinical setting, after adjusting for age and severe KPC-KP infection. Our study confirms the utility of GRS to predict infection risk in patients colonized by KPC-KP.

**IMPORTANCE** The rapid dissemination of carbapenemase-producing Enterobacterales represents a global public health threat. Increased relative load of KPC-producing *Klebsiella pneumoniae* (KPC-KP) within the intestinal microbiota has been associated with an increased risk of bloodstream infection by KPC-KP. We developed a qPCR assay for quantification of the relative KPC-KP intestinal load ($RL_{KPC}$) in 80 colonized patients and examined its association with subsequent all-site KPC-KP infection and all-cause mortality within 90 days. Giannella Risk Score, which predicts infection risk in colonized patients, was independently associated with the development of all-site KPC-KP infection. $RL_{KPC}$ was not associated with all-site KPC-KP infection, possibly reflecting

Address correspondence to Elena Pérez-Nadales, elena.pereznadales@imibic.org, or Julián Torre-Cisneros, julian.torre.sspa@juntadeandalucia.es.

The authors declare no conflict of interest.

the large heterogeneity in patient clinical conditions and infection types. $RL_{KPC}$ was an independent predictor of all-cause mortality within 90 and 30 days in our clinical setting. We hypothesize that KPC-KP load may behave as a surrogate marker for the severity of the patient's clinical condition.

**KEYWORDS** KPC-producing *Klebsiella pneumoniae*, bacterial load, mortality, infection, intestinal colonization

A substantial proportion of patients colonized with *Klebsiella pneumoniae* carbapenemase-producing *K. pneumoniae* (KPC-KP) develop subsequent extraintestinal infections (1–4). In colonized patients, it is possible to define the risk of infection using relevant clinical variables included in validated risk scores such as the Giannella Risk Score (GRS) (5). This score has also been used in combination with the INCREMENT-CPE score (6) to measure the risk of death in infected patients and propose a clinical management algorithm (7, 8).

There is increasing evidence that loss of colonization resistance by the commensal microbiota facilitates the expansion of antibiotic-resistant pathogens (9–11). In previous studies, both low biodiversity and increased relative load of a single organism (elsewhere called "relative abundance" or "intestinal domination") within the gastrointestinal microbiota has been associated with increased risk of infection or death in special populations such as intensive care patients or hematopoietic cell transplant recipients (1, 3, 4, 12–20). It is unknown, however, whether a microbiota-related molecular parameter such as the bacterial load of gut-colonizing antibiotic-resistant bacteria is an adequate predictor for risk of infection or death in more heterogeneous patient populations. To examine this hypothesis, we conducted a prospective observational study of KPC-KP adult carriers with a hospital admission at recruitment or within the 3 prior months in a tertiary teaching hospital with current endemicity by this microorganism.

## RESULTS

**Characteristics of the study cohort.** The main cohort encompassed 80 patients with a median age of 84 years (range 74 to 88), predominantly female (57.5%) (Table 1). Sixty-eight patients (85.0%) were hospitalized at recruitment (approximately 9% in an Intensive Care Unit) and 12 (15%) patients were not hospitalized at recruitment and had a first positive rectal swab in the previous 3 months, according to hospital records. Among hospitalized patients, the median time from hospital admission to the first positive KPC-KP rectal culture was 6 days (range 1 to 14), and the median time to discharge was 17 days (range 9 to 31). Fifteen (18.8%) patients were discharged to a nursing home. The median Charlson score at recruitment was 2 (range 1 to 4). Most patients (78.7%) were classified as rapidly or ultimately fatal according to Mc Cabe's score. The most frequent comorbid conditions were diabetes mellitus (43.8%), chronic renal disease (21.3%), presence of a tumor (17.5%), and recurrent UTI episodes (15.0%). In the previous 3 months, 51.3% patients had a hospitalization episode, 51.3% received proton pump inhibitors, and 16.3% received immunosuppressive therapy. Invasive procedures in the 3 months prior to recruitment included presence of a urinary catheter (70%), surgical procedures (18.8%), nasogastric intubation (22.5%), presence of a central venous catheter (17.5%), mechanical ventilation (13.8%) and surgical procedures (18.8%). Most patients (91.3%) were exposed to antibiotics in the month before recruitment, including amoxicillin-clavulanic acid (23.8%), piperacillin-tazobactam (28.7%), fluoroquinolones (38.8%), cephalosporins (38.8%), carbapenems (10.0%), and aminoglycosides (12.5%). In addition, during the first month of follow-up, exposure to antibiotics was recorded in 85% patients.

To assess the relative KPC-KP intestinal load ($RL_{KPC}$) on rectal swab samples obtained on day 0 of follow-up for the 80 patients, we used a qPCR assay previously validated by our group (21), with some modifications (described in Materials and Methods, Supplementary Methods, and Table S2). The overall median $RL_{KPC}$ was 0.28%, with an interquartile range (range) of 0.001% to 2.70%, and an overall range of <0.001 to 59.39%. We observed significant differences in the distribution of $RL_{KPC}$ values between

**TABLE 1** Clinical characteristics of 80 patients with rectal colonization by KPC-producing *Klebsiella pneumoniae* (KPC-KP)

| Variables | Study cohort<br>$n = 80$ |
|---|---|
| Age (years), median (IQR)[a] | 84 (74 to 88) |
| Male gender | 34 (42.5) |
| | |
| Hospitalization | |
| Hospitalization in the previous 3 months | 41 (51.2) |
| Hospitalization at recruitment | 68 (85.0) |
| Intensive care unit admission | 7 (8.8) |
| Length of hospital stay (days), median (IQR) | 17 (9 to 31) |
| Discharge to a nursing home | 15 (18.8) |
| Hospital readmission during follow-up | 26 (32.5) |
| | |
| Comorbidities | |
| Charlson's index, median (IQR) | 2 (1 to 4) |
| Diabetes mellitus | 35 (43.8) |
| Chronic renal disease | 17 (21.3) |
| Tumor | 14 (17.5) |
| | |
| McCabe score | |
| Nonfatal | 17 (21.3) |
| Rapidly fatal | 32 (40.0) |
| Ultimately fatal | 31 (38.8) |
| | |
| Clinical factors prior to recruitment | |
| Immunosuppressive therapy in the previous 3 month | 13 (16.3) |
| Recurrent urinary tract infections | 12 (15.0) |
| Proton pump inhibitors in the previous month | 41 (51.2) |
| | |
| Invasive procedures and devices in the previous month | 63 (78.8) |
| Urinary catheter | 56 (70.0) |
| Nasogastric intubation | 18 (22.5) |
| Surgical procedures | 15 (18.8) |
| Mechanical ventilation | 11 (13.8) |
| Central venous catheter | 14 (17.5) |
| Endoscopic procedure | 6 (7.5) |
| | |
| Antibiotic exposure in the previous month | 73 (91.3) |
| Amoxicillin/clavulanic acid | 19 (23.8) |
| Piperacillin-tazobactam | 23 (28.7) |
| Ceftazidime-avibactam | 2 (2.5) |
| Cephalosporins | 31 (38.8) |
| Fluoroquinolones | 33 (41.3) |
| Aminoglycosides | 10 (12.5) |
| Carbapenems | 8 (10.0) |
| | |
| Clinical factors during follow-up | |
| Giannella risk score at recruitment, median (IQR) | 5 (5 to 8) |
| Central venous catheter during the first month of follow-up | 9 (11.3) |
| | |
| Antibiotic exposure during the first month of follow-up | 68 (85.0) |
| Amoxicillin/clavulanic acid | 16 (20.0) |
| Piperacillin-tazobactam | 30 (37.5) |
| Ceftazidime-avibactam | 25 (31.1) |
| Cephalosporins | 27 (33.8) |
| Fluoroquinolones | 14 (17.5) |
| Aminoglycosides | 12 (15.0) |
| Carbapenems | 6 (7.5) |
| | |
| $RL_{KPC}$[b] on day 0 of follow-up (%), median (IQR) | 0.28 (0.001 to 2.70) |
| | |
| Clinical outcomes | |
| 90-day all-cause mortality | 33 (41.3) |

**TABLE 1** (Continued)

| Variables | Study cohort<br>n = 80 |
|---|---|
| 30-day all-cause mortality | 22 (27.5) |
| 90-day all-site KPC-KP infection | 33 (41.3) |
| 30-day all-site KPC-KP infection | 32 (40.0) |

[a]IQR, interquartile range.
[b]RL$_{KPC}$, relative load of KPC-KP within the gut microbiota.

68 patients who were hospitalized at recruitment, who showed a median (range) RL$_{KPC}$ of 0.30% (0.01% to 4.82%) and the remaining 12 nonhospitalized patients, who showed a median RL$_{KPC}$ of 0.02% (range 0.002% to 0.27%, $P = 0.048$) (Fig. S4).

The antibiotic susceptibility results for 80 KPC-KP isolates are shown in Table S3. They were mostly resistant to carbapenems (>96.4%), amikacin (95.1%), and fosfomycin (98.8%). The highest sensitivity rates were observed for ceftazidime-avibactam (100% of KPC-KP isolates) and colistin (89.0%). Sensitivity to tigecycline and gentamicin was observed in 35.4% and 26.8% of KPC-KP isolates, respectively.

**Adjusted analysis of the association of the RL$_{KPC}$ with subsequent all-site KPC-KP infection.** All-site KPC-KP infections at day 90 were documented in 33 (41.3%) patients, i.e., nine (11.3%) developed a bloodstream infection (BSI), 13 (16.3%) developed urinary tract infection (UTI), four (5.0%) developed intrabdominal infection (IAI), five (6.3%) developed pneumonia (PNE), and two (2.5%) developed surgical site infection (SSI). KPC-KP all-site infections at day 30 were observed in 32 (40.0%) patients, meaning that the time elapsed from rectal swab collection on day 0 of follow-up to infection onset was below 30 days in all except one patient (who developed an infection on day 89), with a median (range) time of 4 (2 to 7) days. The clinical characteristics and outcomes of the 33 infection episodes are described in Table S4.

The distribution of RL$_{KPC}$ values was not significantly different between patients with no infection (set as reference) versus patients who developed BSI, UTI, PNE, or SSI, and was significantly different in the four patients who developed IAI, with a lower median value ($P = 0.017$, Fig. S5). Receiver operator characteristic (ROC) analysis showed low predicted ability of KPC-KP load for all infection types, including bloodstream infection (Fig. S6).

Follow-up time until all-site infection or all-cause death without an infection, and cumulative incidence functions of the two competing events for the complete follow-up are shown in Fig. S7. Considering all-site KPC-KP infection as the main event and all-cause death as the competing risk, we found the following variables significantly associated with 90-day all-site KPC-KP infection in univariable Fine-Gray regression analysis in the global cohort (Table S5): age (subdistribution hazard ratio or SHR 0.97, 95% confidence interval or CI = 0.95 to 0.99, $P = 0.022$), intensive care unit admission (SHR 3.05, 95% CI = 1.53 to 6.07, $P = 0.002$), length of hospital stay (LOHS) (SHR 1.01, 95% CI = 1.00 to 1.03, $P = 0.039$), surgical procedures (SHR 2.34, 95% CI = 1.17 to 4.70, $P = 0.020$), central venous catheter (SHR 1.98, 95% CI = 0.99 to 3.94, $P = 0.050$) in the previous month, central venous catheter during follow-up (SHR 2.91, 95% CI = 1.38 to 6.13, $P = 0.005$), and Giannella Risk Score (SHR 1.24, 95% CI = 1.17 to 1.32, $P < 0.001$). The same variables were associated with 30-day all-cause infection in univariable analysis, except LOHS (Table S5). In the final multivariable models, the only variable significantly associated with 90-day and 30-day all-site infection was Giannella Risk Score (aSHR 1.23, 95% CI = 1.15 to 1.32, $P < 0.001$ (Table 2A). KPC-KP bacterial load (RL$_{KPC}$) was not found to be associated with time to infection neither in univariable nor in multivariable analysis. When we repeated this analysis in the subcohort of 68 patients who were hospitalized at recruitment, age was significantly associated with subsequent infection, in addition to Giannella Risk Score (Table 2B).

**Adjusted analysis of the association of the RL$_{KPC}$ with all-cause mortality.** All-cause mortality at day 90 was 33/80 (41.3%) with a median (range) time to mortality of 30 (13 to 83) days. All-cause mortality at day 30 was 22/80 (27.5%). Graphs for follow-up time until death or censoring, and survival functions are provided in Fig. S8. We

**TABLE 2** Univariable and multivariable competing risk Fine-Gray regression model for first KPC-KP-related infection episode versus death from other causes (considered competing risk) in the global cohort (A) and in the subcohort patients hospitalized at recruitment (B)

| A | Global cohort (*n* = 80) | | | | |
|---|---|---|---|---|---|
| | Multivariate analysis for 90-day infection[a] | *P* value | | Multivariate analysis for 30-day infection[b] | *P* value |
| | aSHR[e] (95% CI) | | | aSHR (95% CI) | |
| Age (years) | 0.98 (0.95 to 1.00) | 0.092 | | 0.98 (0.95 to 1.00) | 0.069 |
| RL$_{KPC}$[f] on day 0 of follow-up (%) | 0.10 (0.96 to 1.02) | 0.450 | | 0.10 (0.96 to 1.02) | 0.480 |
| Gianella Risk Score | 1.23 (1.15 to 1.32) | <0.001 | | 1.23 (1.15 to 1.32) | <0.001 |
| B | Subcohort of patients hospitalized at recruitment (*n* = 68) | | | | |
| | Multivariate analysis for 90-day infection[c] | *P* value | | Multivariate analysis for 30-day infection[d] | *P* value |
| | aSHR (95% CI) | | | aSHR (95% CI) | |
| Age (years) | 0.97 (0.94 to 0.99) | 0.010 | | 0.97 (0.94 to 0.99) | 0.007 |
| RL$_{KPC}$ on day 0 of follow-up (%) | 0.98 (0.95 to 1.01) | 0.200 | | 0.98 (0.95 to 1.01) | 0.200 |
| Gianella Risk Score | 1.22 (1.14 to 1.32) | <0.001 | | 1.22 (1.14 to 1.32) | <0.001 |

[a]Pseudo likelihood ratio test = 23.1.
[b]Pseudo likelihood ratio test = 23.4.
[c]Pseudo likelihood ratio test = 23.2.
[d]Pseudo likelihood ratio test = 23.4.
[e]aSHR, adjusted subdistribution hazard ratio.
[f]RL$_{KPC}$, relative load of KPC-KP within the gut microbiota.

performed univariable and multivariable Cox regression analyses of variables associated with all-cause mortality at day 90 and 30 (Table S6, S7, respectively) in the global cohort of 80 patients. Dichotomization of the variable LOHS was performed according to ROC curve analyses (Fig. S9). In univariable analysis, variables significantly associated with both 90-day (Table S6) and 30-day (Table S7) all-cause mortality were: RL$_{KPC}$, age, chronic renal disease, ultimately fatal condition according to McCabe score, and INCREMENT-CPE score. In addition, nonfatal McCabe score, immunosuppressive therapy, exposure to aminoglycosides during the first month of follow-up, and developing a bloodstream infection episode were significantly associated with 90-day all-cause mortality in univariable analysis (Table S6), and exposure to fluoroquinolones with 30-day all-cause mortality (Table S7). In our final Cox regression model for 90-day all cause-mortality, stratified by LOHS ≥20 days because this variable did not fulfill the proportional hazards condition, RL$_{KPC}$ was independently associated with the outcome (HR 1.04, 95% CI = 1.01 to 1.07, *P* = 0.008), after adjusting for age (HR 1.05, 95% CI = 1.01 to 1.10, *P* = 0.008) and severe KPC-KP infection (HR 2.20, 95% CI = 0.83 to 5.83, *P* = 0.114) (Table 3A). We obtained

**TABLE 3** Multivariable Cox regression model of factors associated with all cause, 90-day and 30-day mortality in the global cohort of 80 patients (A, B) and in the subcohort of 68 patients hospitalized at recruitment (C, D)

| A | Multivariate model stratified by LOHS[b] ≥ 20 days for 90-day mortality Global cohort (*n* = 80) | | B | Multivariate model stratified by LOHS ≥ 14 days for 30-day mortality Global cohort (*n* = 80) | |
|---|---|---|---|---|---|
| | aHR[d] (95% CI) | *P* value | | aHR (95% CI) | *P* value |
| Age (years) | 1.05 (1.01 to 1.10) | 0.008 | Age (years) | 1.10 (1.03 to 1.18) | 0.004 |
| RL$_{KPC}$[a] on day 0 | 1.04 (1.01 to 1.07) | 0.008 | RL$_{KPC}$ on day 0 | 1.03 (1.00 to 1.06) | 0.027 |
| Severe KPC-KP infection[c] | 2.20 (0.83 to 5.83) | 0.114 | Severe KPC-KP infection | 2.96 (0.97 to 9.07) | 0.057 |
| Likelihood ratio test = 15.2; Wald test = 14.98; Logrank test = 16.86. | | | Likelihood ratio test = 18.12; Wald test = 14.04; Logrank test = 16.97. | | |
| C | Multivariate model stratified by LOHS ≥ 20 days for 90-day mortality Subcohort of hospitalized patients (*n* = 68) | | D | Multivariate model stratified by LOHS ≥ 14 days for 30-day mortality Subcohort of hospitalized patients (*n* = 68) | |
| | aHR (95% CI) | *P* value | | aHR (95% CI) | *P* value |
| Age (years) | 1.06 (1.02 to 1.11) | 0.003 | Age (years) | 1.12 (1.04 to 1.20) | 0.002 |
| RL$_{KPC}$ on day 0 | 1.04 (1.01 to 1.07) | 0.010 | RL$_{KPC}$ on day 0 | 1.03 (1.00 to 1.06) | 0.032 |
| Severe KPC-KP infection | 4.06 (1.44 to 11.47) | 0.008 | Severe KPC-KP infection | 4.95 (1.51 to 16.22) | 0.008 |
| Likelihood ratio test = 19.28 Wald test = 17.67; Logrank test = 20.3. | | | Likelihood ratio test = 20.9 Wald test = 15.9; Logrank test = 19.07. | | |

[a]RL$_{KPC}$, relative load of KPC-KP within the gut microbiota.
[b]LOHS, Length of hospital stay.
[c]Severe KPC-KP infection was considered in patients with infections and an INCREMENT-CPE SCORE >7 (6, 7).
[d]aHR, adjusted hazard ratio.

similar findings in a multivariable Cox model for 30-day mortality, stratified by LOHS $\geq$14 days, including $RL_{KPC}$ (HR 1.03, 95% CI = 1.00 to 1.06, $P$ = 0.027), age (HR 1.10, 95% CI = 1.03 to 1.18, $P$ = 0.004), and severe KPC-KP infection (HR 2.96, 95% CI = 0.97 to 9.07, $P$ = 0.057) (Table 3B). Next, we repeated these analyses in the subcohort of 68 hospitalized patients. This confirmed the association of $RL_{KPC}$ with 90-day mortality in this subcohort (HR 1.04, 95% CI = 1.01 to 1.07, $P$ = 0.010), after stratifying by LOHS $\geq$20 days, and adjusting for age (HR 1.06, 95% CI = 1.02 to 1.11, $P$ = 0.003) and severe KPC-KP infection (HR 4.06, 95% CI = 1.44 to 11.47, $P$ = 0.008) (Table 3C). In the case of 30-day all-cause mortality, a Cox model in the subcohort of hospitalized patients, stratified by LOHS $\geq$14 days, included $RL_{KPC}$ (HR 1.03, 95% CI = 1.00 to 1.06, $P$ = 0.032), age (HR 1.12, 95% CI = 1.04 to 1.20, $P$ = 0.002), and severe KPC-KP infection (HR 4.95, 95% CI = 1.51 t0 16.22, $P$ = 0.008) (Table 3D). All variables included in the final stratified models fulfilled the proportional hazard assumption (Fig. S10).

## DISCUSSION

Our study investigates the association between the relative intestinal load of KPC-KP, determined by qPCR, and the risk of subsequent all-site KPC-KP infection and all-cause death in patients with a recent diagnosis of KPC-KP intestinal colonization. A previous study has confirmed an association between KPC-KP intestinal load and bacteremia (18). Our study investigates the association with any type of KPC-KP infection. However, we have not been able to verify this association. In the multivariable analysis, Giannella Risk Score showed an association with the risk of infection and it can be considered that it is currently a valid option to determine this risk in carbapenem-resistant *Enterobacterales* (CRE)-colonized patients (5, 7, 8). Several explanations could justify the lack of association of KPC-KP intestinal load with subsequent infection in our study. First, we have a highly heterogeneous patient population in terms of baseline clinical conditions as well as infection types. Sun et al. showed that increased *Klebsiella* intestinal load (>22%) was significantly associated with subsequent all-site infections and with BSI but not with urinary tract and respiratory infections in an ICU setting (22). Second, we obtained a single determination of KPC-KP load at recruitment, so it is possible that we may have missed the highest relative load values prior to infection. Shimasaki and collaborators reported that a KPC-KP intestinal load cutoff of 22% predicted KPC-KP bacteremia in colonized patients according to ROC curves; however, they used for each patient the rectal swab sample with the highest load of KPC-KP during follow-up (18). Interestingly, graphical data showing the weekly fluctuation in KPC-KP load in 11 patients who developed infections in the Shimasaki study highlighted that peak KPC-KP load values were usually detected in the week prior to onset of the BSI episode (18). Taur and colleagues reported that enteric domination by Enterobacterales ($\geq$30% of the microbiota) is associated with significant increase in BSI risk in patients undergoing allogeneic hematopoietic stem cell transplantation (allo-HCT), with a median time between domination and bacteremia of 7 days (15). In a different study of 708 allo-HCT patients, Stoma and collaborators showed that intestinal domination ($\geq$30%) by Gram-negative bacteria was highly predictive of Gram-negative BSI however, they collected fecal specimens every 3 to 5 days in subjects and observed instances where intestinal domination was short-lived and only lasted 1 day (13). Research from our group has shown that the time from first detection of KPC-KP colonization, which typically occurs during a hospitalization episode, plays a role in infection risk, with 80% of patients showing infection onset within 15 days of first diagnosis (23). In our present study, most patients (all except one) developed infections within this 15-day time window. When we performed a subanalysis with the subset of 68 patients hospitalized at recruitment, this parameter was still not predictive of infection. More detailed knowledge of the timing when major shifts in intestinal bacterial communities take place under different clinical settings and its short- and long-term health consequences is needed.

Previous evidence from our group has shown that selective intestinal decontamination reduces the risk of crude mortality in patients colonized by KPC-KP (24). We have

also provided evidence that KPC-KP rectal colonization is not associated *per se* with crude mortality (25). Instead, mortality increases when colonized patients develop severe KPC-KP infection, i.e., INCREMENT-CPE mortality score (ICS) >7 (25). In our COX regression model, after stratifying by LOHS and adjusting by age, and ICS > 7, KPC-KP load was significantly associated with 90-day and 30-day mortality. Importantly, our cohort includes an aging population with many comorbidities, and 10% (8/80) of patients developed severe KPC-KP infections. Moreover, attributable mortality was 17.5% (14/80), compared with 41.2% (33/80) overall mortality. Therefore, we may consider that KPC-KP relative intestinal load may also be associated with noninfectious mortality. Age-related alterations in the gut microbiome are influenced by factors such as progressive physiological deterioration, and lifestyle-linked factors, including diet, medication, and reduced social contact (26). In people over the age of 70, a study reported a decrease in anaerobic bacteria such as *Bifidobacterium* spp., which has a role in the stimulation of the immune system and metabolic processes, and an increase in *Clostridium* and Proteobacteria (27). Hospitalization may exacerbate microbiota dysbiosis in frail, older people as a result of medication and exposure to health care invasive procedures. The niches left open within the intestinal microbiota may be occupied by overgrowth of microorganisms that have pathogenic potential at high densities, with the health care personnel acting as a transmission vector in the hospital environment. Members of the Proteobacteria, such as *Escherichia coli* or *K. pneumoniae*, normally represent less than 2% of the microbiota, but in a dysbiotic state these bacteria can represent up to 30% of the total species (18). We hypothesize that an increased intestinal load of KPC-KP within the gut microbiota in our clinical context may behave as a surrogate marker of debilitated health condition because of frailty, higher burden of comorbidities, or age-related alterations in the gut microbiome and immune dysfunction (immune senescence), which may in turn lead to higher risk of infection or death.

The main limitation of this study was the relatively small number of cases ($n = 80$), which limits our ability to investigate associations by site of infection. In addition to this, an important limitation is not having monitored KPC-KP load as a time-dependent variable during follow-up. There are also limitations inherent to our qPCR approach for estimation of KPC-KP relative load in rectal swabs because this method does not account for bias derived from the potential presence of dead bacterial cells in rectal swabs or differences in target ($bla_{KPC}$) and internal reference (16SrRNA) gene copy number between different bacterial cells or bacterial species. As an alternative, some studies have employed rRNA sequencing (18, 22), which has the advantage of providing informative data about microbiota diversity. However, compared with rRNA sequencing, qPCR assays have the advantage that could be more readily translated into clinical practice because many clinical microbiology laboratories are proficient in the implementation of these assays.

In summary, our study supports an association of KPC-KP intestinal load with mortality but not with subsequent infection. Patients may become colonized upon hospital admission or health care-related procedures because the colonizing KPC-KP strain encounters an altered microbiota incapable of stopping colonization "immunologically" (9–11, 28). Based on our present and previous findings (23), we hypothesize that the 15-day time window following initial acquisition of CRE during hospitalization may be critical for successful intestinal propagation and propose to design preventive strategies in specific high-risk groups of patients, which may include monitorization of microbiota-related parameters (i.e., CRE intestinal load, microbial diversity) in order to properly assess their potential as biomarkers for adverse outcomes. Once colonized, the risk of infection should be estimated by classical clinical variables included in the Giannella Risk Score (this study, [5, 7, 8]) while INCREMENT-CPE score is clinically confirmed as a good predictor for death in patients who develop KPC-KP infections (6–8, 25).

## MATERIALS AND METHODS

**Study design.** The study was a prospective, observational, cohort study (KLEBCOM, PI16/01631, FIS-ISCIII) conducted at Reina Sofía University Hospital (Cordoba, Spain), a 1,000-bed tertiary University

teaching hospital with endemicity by a *Klebsiella pneumoniae* carbapenemase-producing *Klebsiella pneumoniae* (KPC-KP) clone of sequence type (ST) 512, following an initial outbreak in 2012 (29). The study recruitment period was from January 2018 to February 2019. Inclusion criteria were adult patients with a first KPC-KP-positive rectal surveillance culture during a hospital admission throughout the recruitment period or within the 3 prior months (November to December 2018). During the study period, KPC-KP rectal colonization screening was systematically performed in our hospital by means of a rectal swab culture in patients admitted to high-risk units (intensive care unit and hematology unit) and those undergoing abdominal surgery or transplants. In addition, a colonization study may have been requested by the clinician for various reasons: previous admission to high-risk units, origin of the patient from a health care center, or sharing a room with colonized or infected patients. Subjects with a first KPC-KP colonization during a hospital stay in the 3 months prior to start of recruitment were identified through review of hospital records, contacted by phone, and invited to participate in the study. Following the signing of the informed consent, a rectal swab sample was obtained for confirmation of KPC-KP colonization and quantification of KPC-KP intestinal relative load (baseline $RL_{KPC}$) in all patients and this was considered the start date of follow-up (day 0). Only those patients with a positive KPC-KP culture on day 0 of follow-up were included in the study, while uncolonized patients were excluded. In addition, colonized patients who received a clinical indication of selective intestinal decolonization with oral nonadsorbable antibiotics during follow-up were excluded from the present analysis and they are described elsewhere (21). All patients were followed until 90 days or death. Patients were seen at the hospital monthly for clinical and microbiological follow-up by means of rectal swab screening. In the case of patients with reduced mobility, a dedicated nurse went to the patient's residence for clinical and microbiological follow-up visits.

The study protocol was approved by the Ethics Committee of Reina Sofía University Hospital, Córdoba, Spain (Code: KLEBCOM study, Act 253, ref. 3197). Informed consent was obtained from each patient or next of kin in accordance with the principles of the Declaration of Helsinki. All patient data were anonymized. This report follows STROBE recommendations (Table S1) (30).

**Variables and definitions.** Rectal KPC-KP colonization was defined as the isolation of KPC-KP in a rectal swab in the absence of clinical signs and symptoms of infection. All KPC-KP infection episodes were microbiologically confirmed and were defined according to the criteria established by the guidelines of the Centers for Disease Control and Prevention (31). The relative intestinal load of KPC-KP within the gut microbiota ($RL_{KPC}$) was defined as the percentage of the number of copies of the $bla_{KPC}$ gene (representing KPC-KP) relative to 16S rRNA genes (representing total bacteria) estimated in total genomic DNA extracted from a patient's rectal swab obtained on day 0 of follow-up, according to qPCR analysis (see below).

We analyzed two outcomes at 30 and 90 days: (i) development of KPC-KP all-site infections and (ii) all-cause mortality. Exposure variables included demographics, hospitalization-related variables, immunosuppressive therapy, comorbid conditions and comorbidity severity indexes, i.e., Charlson index (32), McCabe's score (33), diabetes mellitus, chronic renal disease, recurrent urinary tract infection, tumor and solid organ transplantation, invasive procedures, exposure to systemic antibiotics, and Giannella Risk Score, which predicts the risk of infection in patients colonized by carbapenem-resistant Enterobacterales (5, 7, 8).

**Quantification of the relative intestinal load of KPC-KP in rectal swabs.** Rectal swab samples were collected on day 0 of follow-up using eSwab rectal swabs (Copan, Brescia, Italy), which contain liquid Amies transport medium, and were submitted both to culture-based and qPCR-based quantification of the relative intestinal load of KPC-KP, based on the procedures originally published by Lerner et al. (34) (summarized in Fig. S1). In addition, microbial identification was performed for all CRE isolated from rectal swab cultures following standard microbiological procedures. Both assays (culture and qPCR) are described in detail in Supplementary Methods and Table S2. We found a good correlation of the KPC-KP relative intestinal load values obtained by both methods in 80 baseline rectal swab samples, according to nonparametric Kendall's rank correlation (tau test 0.495, *P*-value = <0.001) (Fig. S2).

**Microbiological methods.** Identification and antimicrobial susceptibility testing of *K. pneumoniae* isolates were routinely performed using two commercial microdilution methods: MicroScan Walk-Away Gram-negative NC53 pane (Beckman Coulter, USA) and/or Sensititre Antimicrobial Susceptibility Testing System (Thermo Fisher, USA). For gentamicin and fosfomycin, a broth microdilution method was performed using *E. coli* (ATCC 25922) and *Pseudomonas aeruginosa* (ATCC 27853) as control strains. Susceptibility and resistance categories were assigned according to EUCAST breakpoints (35). In some cases, the identification was confirmed by MALDI-TOF (MALDI Biotyper, Bruker Daltonics GmbH, USA). KPC production was confirmed by immunochromatography with NG-test CARBA 5 (NG Biotech, France) and presence of the $bla_{KPC}$ gene was confirmed by qPCR analysis.

**Data analysis.** Categorical variables were evaluated by using the $\chi^2$-test or Fisher's exact test, as appropriate. Continuous variables were compared using Student's t or Mann–Whitney U test. For continuous variables dichotomization was performed according to the optimal cut-off point determined using ROC curves analysis for each specific outcome, and if considered of clinical relevance. All *P*-values were two-tailed, and a *P*-value <0.05 was considered statistically significant.

We used survival analysis to study the association of variables with study outcomes. To examine variables associated with development of all-site KPC-KP infection, we used competing-risk survival regression analysis, based on the Fine and Gray function, and considering death before KPC-KP infection as a competing risk. Cumulative incidence functions for each event ("infection" versus "death before infection") were generated, using the $\chi^2$-test to compare among categories for each event. To examine variables associated with all-cause mortality, we explored categorical variables with the Kaplan Meier survival function and log-rank test. Crude and adjusted Cox regression hazard ratios with their 95% confidence intervals were calculated to measure time to study outcomes (90-day and 30-day mortality), and fulfillment of the proportional hazard assumption was performed with Schoenfeld residual test. The selection

of variables in our multivariable models was further supported by performing the ranking of all explanatory (independent) variables included in the KLEBCOM database using the random survival forest score of importance (36), and considering the top 10 variables (Fig. S3). Statistics were performed with R software version 4.0.4. Statistics were performed with R software version 4.0.4.

## SUPPLEMENTAL MATERIAL

Supplemental material is available online only.
**SUPPLEMENTAL FILE 1**, PDF file, 1.7 MB.

## ACKNOWLEDGMENTS

This work was supported by: (i) Plan Estatal de I+D+I 2013-2016, cofinanced by the ISCIII-Subdirección General de Evaluación y Fomento de la Investigación and the Fondo Europeo de Desarrollo Regional-FEDER (FIS PI16/01631-KLEBCOM granted to E.P.N.); (ii) Plan Nacional de I+D+i 2013-2016 and Instituto de Salud Carlos III (ISCIII), Subdirección General de Redes y Centros de Investigación Cooperativa, Ministerio de Ciencia, Innovación y Universidades, and cofinanced by the European Development Regional Fund "A Way to Achieve Europe" Operative Program Smart Growth 2014–2020, Spanish Network for Research in Infectious Diseases (REIPI) (RD16/0016/0008); and (iii) Consejería de Salud y Familias, Junta de Andalucía (RH-0065-2020 granted to E.P.N.).

J.T.C. has served as scientific advisor for a research/consensus project for Pfizer, as an expert in a consensus document for InfectoPharm, and has received payment for lectures, including service on speaker bureaus, and for the development of educational presentations for Pfizer, AstraZeneca, Shionogi, and Merck. A.C. has received honoraria for the development of educational presentations for Pfizer and Shionogi. L.M.M. has been a consultant for MSD and Shionogi, has served as a speaker for MSD, Becton, Dickinson, Pfizer, and Shionogi, and has received research support from Pfizer and Janssen. All other authors declare no potential conflicts. All authors have submitted the ICMJE Form for Disclosure of Potential Conflicts of Interest. Conflicts that the editors consider relevant to the content of the manuscript have been disclosed.

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
