## [Reviewer comments · Microbiology Spectrum]

Microbiology Spectrum

Prognostic significance of the relative load of KPC-producing *Klebsiella pneumoniae* within the intestinal microbiota in a prospective cohort of colonized patients.

Elena Pérez-Nadales, Alejandra M. Natera, Manuel Recio-Rufián, Julia Guzmán-Puche, Juan Antonio Marín-Sanz, Carlos Martín-Pérez, Ángela Cano, Juan José Castón, Cristina Elías-López, Isabel Machuca, Belén Gutiérrez-Gutiérrez, Luis Martínez-Martínez, and Julian Torre-Cisneros

Corresponding Author(s): Elena Pérez-Nadales, Maimonides Biomedical Research Institute of Cordoba, Reina Sofía University Hospital, University of Cordoba (IMIBIC/HURS/UCO)

Review Timeline:

Submission Date:	January 25, 2022
Editorial Decision:	February 27, 2022
Revision Received:	May 4, 2022
Accepted:	May 20, 2022

Editor: Ahmed Babiker

Reviewer(s): The reviewers have opted to remain anonymous.

Transaction Report:

DOI: <https://doi.org/10.1128/spectrum.02728-21>

February 27, 2022

Dr. Elena Pérez-Nadales
Maimonides Biomedical Research Institute of Cordoba, Reina Sofía University Hospital, University of Cordoba
(IMIBIC/HURS/UCO)
Córdoba
Spain

Re: Spectrum02728-21 (Prognostic significance of the relative load of *Klebsiella pneumoniae* carbapenemase-producing *Klebsiella pneumoniae* within the intestinal microbiota in a prospective cohort of colonized patients.)

Dear Dr. Elena Pérez-Nadales:

Thank you for submitting your manuscript to Microbiology Spectrum. I would be happy to consider a revised version of the manuscript which addresses the reviewers comments, with which I agree. When submitting the revised version of your paper, please provide (1) point-by-point responses to the issues raised by the reviewers as file type "Response to Reviewers," not in your cover letter, and (2) a PDF file that indicates the changes from the original submission (by highlighting or underlining the changes) as file type "Marked Up Manuscript - For Review Only". Please use this link to submit your revised manuscript - we strongly recommend that you submit your paper within the next 60 days or reach out to me. Detailed instructions on submitting your revised paper are below.

Link Not Available

Sincerely,

Ahmed Babiker

Journals Department
Reviewer comments:

Reviewer #1 (Comments for the Author):

Summary:

The submitted article provides a unique look at the significance of bacterial load of KPC-producing *Klebsiella pneumoniae* in predicting subsequent infection or death. The authors developed a PCR method to provide a relative value of bla-KPC genes. While the bacterial load did not correlate with infection, it was independently correlated with death. The Giannella Risk Score was associated with subsequent infection.

Major comments:

Line 133: The number of copies of the bla-KPC gene does not necessarily mean that these were all linked to *K. pneumoniae*. Other Enterobacteriaceae could harbor this gene. How can you be confident that you are classifying a single species' abundance? It's unclear from the supplementary materials if an additional identification step was performed prior to PCR.

For patients who were retrospectively identified to have KPC-KP colonization, was there any verification they were still colonized on enrollment? Given that colonization can be transitory, were any subsequent cultures collected on the participants?

Did you collect any information on other outcome metrics, such as total infections? If, as you propose, KPC-KP colonization signifies dysbiosis, bacterial translocation could occur from any species, not just KP (see: <https://www.ncbi.nlm.nih.gov/pmc/articles/PMC7193040/>). This would be especially interesting to know given the relatively low loads of KPCs in your study.

Line 362-365: Why does colonization increase mortality, independent of infection? I'm not sure I find the argument that it represents a decompensation of underlying disease cogent.

Minor comments:

The manuscript would benefit from spellcheck.

Line 83: Do you mean surrogate instead of subrogate?

Line 112: How is endemicity defined? Is this based on colonization screening or isolates from clinical infections?

Line 115: When you say "first diagnosis" do you mean the individuals may have been colonized for an indefinite period of time? How were the patients selected (beyond colonization status)? Was it just a convenience sample of the first 80 patients identified? Where in the hospital were they located (surgical, medical, all wards)? Is Day 0 the first day of admission, the first day with a positive swab, or the start of the study (if it was cross-sectional)? It would be helpful to know more about the study design for the purposes of generalizability.

Why did you stratify 30-day mortality by LOH >14 days and 90-day mortality by LOH >20 days? Were there clear splines in the data?

Hospitalization at recruitment may not be a significant factor in your analysis because of the small number of people who were not hospitalized. I would include this as a limitation.

I'm surprised that previous hospitalization was not considered in the list of risk factors. Patients who are high utilizers of healthcare might be expected to have higher mortality in general, but this was not accounted for in the analysis.

Supplementary Table 7: There are two columns listed as multivariable analysis. Presumably the first one is univariable analysis.

Reviewer #2 (Comments for the Author):

Authors studied bacterial load of KPC producing *Klebsiella pneumoniae* within the intestinal microbiota in a prospective cohort of colonized patients. They also analyze the prognosis of this value. Bacteria load unlike other studies did not influence in latter infection. A qPCR assay was used for the study of bacterial load. The statistical analysis is sound

General comments

The inclusion criteria were patients with a first diagnosis of intestinal KPC-KP colonization during a hospital admission at the time of recruitment or within the three prior months. It seems different groups for the analysis. Do the authors perform a separate analysis in both groups?

How do the authors control the sampling (rectal swabs and not perirectal swabs?) Include type and/or trade mark of rectal swabs.

This group has experience with intestinal decontamination (reference 34). Was the use of this procedure considered an exclusion criteria?

The total number of patients is very low and this might have bias the results. Nevertheless the did a rigorous analysis of the data obtained.

Specific comments

Title. I will recommend to use the acronyms KPC instead "Klebsiella pneumoniae carbapenemase" This will make the title more readable.

Line 54. Is this a specific 16S rRNA for *K. pneumoniae* or total 16S rRNA?

Line 84. Immunological conditions? This might be out of the scope

Line 193. Range of patient's age is very high. Were younger patients excluded?

Line 223. Antibiotic susceptibility instead antibiotic sensitivity

Line 227. There is no EUCAST criteria for tigecycline and *K. pneumoniae*. Do the authors used the PK-PD breakpoint for this antibiotic?

Line 236. Was the antibiotic treatment influence this figure?

Line 363-365. This can be due to the progressive elimination of Firmicutes and increase of Proteobacteria with aging Table S3. Include S / I / R in this order.

Staff Comments:

Preparing Revision Guidelines

Please return the manuscript within 60 days; if you cannot complete the modification within this time period, please contact me. If you do not wish to modify the manuscript and prefer to submit it to another journal, please notify me of your decision immediately so that the manuscript may be formally withdrawn from consideration by Microbiology Spectrum.

Microbiology Spectrum

Review of “Prognostic significance of the relative load of *Klebsiella pneumoniae* carbapenemase-producing *Klebsiella pneumoniae* within the intestinal microbiota in a prospective cohort of colonized patients.”

Summary:

The submitted article provides a unique look at the significance of bacterial load of KPC-producing *Klebsiella pneumoniae* in predicting subsequent infection or death. The authors developed a PCR method to provide a relative value of bla-KPC genes. While the bacterial load did not correlate with infection, it was independently correlated with death. The Giannella Risk Score was associated with subsequent infection.

Major comments:

Line 133: The number of copies of the bla-KPC gene does not necessarily mean that these were all linked to *K. pneumoniae*. Other Enterobacteriaceae could harbor this gene. How can you be confident that you are classifying a single species' abundance? It's unclear from the supplementary materials if an additional identification step was performed prior to PCR.

For patients who were retrospectively identified to have KPC-KP colonization, was there any verification they were still colonized on enrollment? Given that colonization can be transitory, were any subsequent cultures collected on the participants?

Did you collect any information on other outcome metrics, such as total infections? If, as you propose, KPC-KP colonization signifies dysbiosis, bacterial translocation could occur from any species, not just KP (see: <https://www.ncbi.nlm.nih.gov/pmc/articles/PMC7193040/>). This would be especially interesting to know given the relatively low loads of KPCs in your study.

Line 362-365: Why does colonization increase mortality, independent of infection? I'm not sure I find the argument that it represents a decompensation of underlying disease cogent.

Minor comments:

The manuscript would benefit from spellcheck.

Line 83: Do you mean surrogate instead of subrogate?

Line 112: How is endemicity defined? Is this based on colonization screening or isolates from clinical infections?

Line 115: When you say “first diagnosis” do you mean the individuals may have been colonized for an indefinite period of time? How were the patients selected (beyond colonization status)? Was it just a convenience sample of the first 80 patients identified? Where in the hospital were they located (surgical, medical, all wards)? Is Day 0 the first day of admission, the first day with a positive swab, or the start of the study (if it was cross-sectional)? It would be helpful to know more about the study design for the purposes of generalizability.

Why did you stratify 30-day mortality by LOH >14 days and 90-day mortality by LOH >20 days? Were there clear splines in the data?

Hospitalization at recruitment may not be a significant factor in your analysis because of the small number of people who were not hospitalized. I would include this as a limitation.

I'm surprised that previous hospitalization was not considered in the list of risk factors. Patients who are high utilizers of healthcare might be expected to have higher mortality in general, but this was not accounted for in the analysis.

Supplementary Table 7: There are two columns listed as multivariable analysis. Presumably the first one is univariable analysis.

Infectious Diseases Group (GC-03)
Maimonides Biomedical Research Institute of Cordoba
Edificio IMIBIC, Avda. Menéndez Pidal s/n. Postal code: 14004, Córdoba, Spain

Cordoba, April 14th, 2022

Response to reviewer comments for the article entitled "**Prognostic significance of the relative load of *Klebsiella pneumoniae* carbapenemase-producing *Klebsiella pneumoniae* within the intestinal microbiota in a prospective cohort of colonized patients**" (Spectrum02728-21)" to be considered for publication in *Microbiology Spectrum* as "Original Article".

Reviewer #1 (Comments for the Author):

Summary:

The submitted article provides a unique look at the significance of bacterial load of KPC-producing *Klebsiella pneumoniae* in predicting subsequent infection or death. The authors developed a PCR method to provide a relative value of bla-KPC genes. While the bacterial load did not correlate with infection, it was independently correlated with death. The Giannella Risk Score was associated with subsequent infection.

Major comments:

Major comment 1: Line 133: The number of copies of the bla-KPC gene does not necessarily mean that these were all linked to *K. pneumoniae*. Other Enterobacteriaceae could harbour this gene. How can you be confident that you are classifying a single species' abundance? It's unclear from the supplementary materials if an additional identification step was performed prior to PCR.

Response: For all rectal swabs, quantification of bacterial load was performed by two methods in parallel: qPCR and culture on selective growth medium. This was followed by identification of any Enterobacteriales growing in CRE-selective culture medium, following standard microbiological procedures. We have re-written the main Methods section to make this clearer. In addition, we have included a new **Supplementary Figure S1** (also included below) summarizing the processing of rectal swabs in our study, which highlights all molecular and microbiological tests performed for each sample.

Major comment 2: For patients who were retrospectively identified to have KPC-KP colonization, was there any verification they were still colonized on enrolment? Given that colonization can be transitory, were any subsequent cultures collected on the participants?

Response: Regarding the first question, yes, following signing of the informed consent by patients who were retrospectively identified, a rectal swab sample was obtained for confirmation of colonization persistence, considered as day 0 of follow-up. Among these patients, only those with a positive KPC-KP culture on day 0 were included in the study, while uncolonized patients were not included in our cohort.

Regarding the second question, the answer is yes. In this prospective cohort study, all patients were cited for monthly visits for clinical and microbiological follow-up. In the case of patients with reduced mobility discharged to a residential care or nursing home, a dedicated nurse travelled to the patient's residence to perform the monthly visits during the study period. In the present paper we focus on the impact of baseline KPC-KP relative intestinal load (day 0) on infection and mortality within 3 months of enrolment. Due to space limitations, results on the natural history of KPC-KP colonization, as well as progression of KPC-KP bacterial load kinetics and risk factors associated with eradication of KPC-KP in our cohort study are the focus of a different analysis and paper.

All this information has been clarified in Methods, as follows:

Lines 117-148 (Marked copy): The study recruitment period was from January 2018 to February 2019. Inclusion criteria were adult patients with a first KPC-KP-positive rectal surveillance culture during a hospital admission throughout the recruitment period or within the three prior months (November-December 2018). During the study period, KPC-KP rectal colonization screening was systematically performed in our hospital by means of a rectal swab culture in patients admitted to high-risk units (intensive care unit and hematology unit) and those undergoing abdominal surgery or transplants. In addition, a colonization study may have been requested by the clinician for various reasons: previous admission to high-risk units, origin of the patient from a healthcare center, or sharing a room with colonized or infected patients. Subjects with a first KPC-KP colonization during a hospital stay in the three months prior to start of recruitment were identified through review of hospital

records, contacted by phone and invited to participate in the study. Following the signing of the informed consent, a rectal swab sample was obtained for confirmation of KPC-KP colonization and quantification of KPC-KP intestinal relative load (baseline RL_{KPC}) in all patients and this was considered the start date of follow-up (day 0). Only those patients with a positive KPC-KP culture on day 0 of follow-up were included in the study, while uncolonized patients were excluded. In addition, colonized patients who received a clinical indication of selective intestinal decolonization with oral non-adsorbable antibiotics during follow-up were excluded from the present analysis and they are described elsewhere (23). All patients were followed until 90 days or death. Patients were seen at the hospital on a monthly basis for clinical and microbiological follow-up by means of rectal swab screening. In the case of patients with reduced mobility, a dedicated nurse went to the patient's residence for clinical and microbiological follow-up visits.

Major comment 3: Did you collect any information on other outcome metrics, such as total infections? If, as you propose, KPC-KP colonization signifies dysbiosis, bacterial translocation could occur from any species, not just KP (see: <https://www.ncbi.nlm.nih.gov/pmc/articles/PMC7193040/>). This would be especially interesting to know given the relatively low loads of KPCs in your study.

Response: This is an interesting hypothesis, which unfortunately was not studied in our work. Our study focused on KPC-KP infections. Unfortunately, data on the total number of infections was not recorded during data collection. As the reviewer suggests, during a state of dysbiosis, niches left open after microbiota alteration may be occupied by overgrowth of microorganisms that have pathogenic potential at high densities. Members of the Proteobacteria, such as *Escherichia coli* or *K. pneumoniae*, normally represent < 2% of the microbiota, but in a dysbiotic state these bacteria can represent up to 30% of the total species. On the other hand, with the healthcare personnel acting as a transmission vector, this ecological gap may be filled by other environmental opportunistic pathogens that frequently carry genetic determinants of resistance in the hospital setting. It would be of interest to analyze this hypothesis in future studies in our hospital.

Major comment 4: Line 362-365: Why does colonization increase mortality, independent of infection? I'm not sure I find the argument that it represents a decompensation of underlying disease cogent.

Response: We agree with the reviewer. To keep the word limit, we oversimplified our discussion of this point. We have re-written this part of the discussion considering major comments 3 and 4, as follows:

~~Before: We hypothesize that the relative intestinal load of KPC-KP may behave as a surrogate marker for decompensation of the underlying disease, comorbidities or patient's immune status (immunosenescence) in our elderly patient population, which ultimately may lead to patient's death.~~

Now (Lines 411-428, marked copy): Age-related alterations in the gut microbiome are influenced by factors such as progressive physiological deterioration, and lifestyle-linked factors including diet, medication and reduced social contact (38). In people over the age of 70, a study reported a decrease in anaerobic bacteria such as *Bifidobacterium* spp., which has a role in the stimulation of the immune system and metabolic processes, and an increase in *Clostridium* and Proteobacteria (39). Hospitalization may exacerbate microbiota dysbiosis in frail, older people as a result of

medication and exposure to healthcare invasive procedures. The niches left open within the intestinal microbiota may be occupied by overgrowth of microorganisms that have pathogenic potential at high densities, with the healthcare personnel acting as a transmission vector in the hospital environment. Members of the Proteobacteria, such as *Escherichia coli* or *K. pneumoniae*, normally represent less than 2% of the microbiota, but in a dysbiotic state these bacteria can represent up to 30% of the total species (18). We hypothesize that an increased intestinal load of KPC-KP within the gut microbiota in our clinical context may behave as a surrogate marker of debilitated health condition because of frailty, higher burden of comorbidities, or age-related alterations in the gut microbiome and immune dysfunction (immune senescence), which may in turn lead to higher risk of infection or death.

Minor comments:

Minor comment 1: The manuscript would benefit from spellcheck.

Response: Thank you. We have reviewed the manuscript thoroughly to correct any spelling errors.

Minor comment 2: Line 83: Do you mean surrogate instead of subrogate?

Response: Corrected, thank you.

Minor comment 3: Line 112: How is endemicity defined? Is this based on colonization screening or isolates from clinical infections?

Response: After controlling an initial outbreak by a KPC3-KP ST512 clone in the summer of 2012, endemicity was reached in our hospital and has prevailed to date, with cases of colonisation and infection with no epidemiological relationship between them (and therefore without epidemiological outbreak criteria). In high-risk units (ICU, haematology) this criterion is based on colonisation screening. In the rest of the hospital units, it is based on isolations of clinical infections. This general information has been included in Methods, "Study design" section (Lines 113-148, marked copy).

Minor comment 4: Line 115: When you say "first diagnosis" do you mean the individuals may have been colonized for an indefinite period of time? How were the patients selected (beyond colonization status)? Was it just a convenience sample of the first 80 patients identified? Where in the hospital were they located (surgical, medical, all wards)? Is Day 0 the first day of admission, the first day with a positive swab, or the start of the study (if it was cross-sectional)? It would be helpful to know more about the study design for the purposes of generalizability.

Response: We thank the reviewer for highlighting the need to expand and clarify the information on study design in the main manuscript. As explained in our response to "Major comment 2", we have rewritten the Methods section accordingly. Thus, details regarding study design, selection of patients, and criteria for rectal swab surveillance in our hospital during the study period are now explained in detail in Methods, "Study design" section (Lines 113-148, marked copy).

Minor comment 5: Why did you stratify 30-day mortality by LOH >14 days and 90-day mortality by LOH >20 days? Were there clear splines in the data?

Response: Dichotomization of the variable length of hospital stay (LOHS) was performed according to ROC curve analyses. This explanation has been included in the text (Lines 319-320, marked copy), and the ROC curves are shown in **Supplementary Figure S9**.

Minor comment 6: Hospitalization at recruitment may not be a significant factor in your analysis because of the small number of people who were not hospitalized. I would include this as a limitation.

I'm surprised that previous hospitalization was not considered in the list of risk factors. Patients who are high utilizers of healthcare might be expected to have higher mortality in general, but this was not accounted for in the analysis.

Response: Thank you. This variable was included in our analysis as "Hospitalization in the previous three months" (please, see **Table 1**). We explored its impact on infection and mortality but found no independent association in our multivariable models. Moreover, in our work, we conducted a first exploratory analysis based on survival Random Forest to identify variables with a strong relative importance for survival. This analysis was previously not mentioned in the manuscript's methods due to space limitations, however we consider it of interest for the reader and have now included a brief description in Statistical Methods (Lines 226-231, marked copy). In addition, we have also included a new **Supplementary Figure S2**, which summarizes the results of this random survival forest analysis. As shown in this **Figure S2**, the variable "Hospitalization in the previous three months" showed negative variable importance with regards to the outcome.

Minor comment 8: Supplementary Table 7: There are two columns listed as multivariable analysis. Presumably the first one is univariable analysis.

Response: Corrected, thank you.

Reviewer #2 (Comments for the Author):

Authors studied bacterial load of KPC producing *Klebsiella pneumoniae* within the intestinal microbiota in a prospective cohort of colonized patients. They also analyse the prognosis of this value. Bacteria load unlike other studies did not influence in latter infection. A qPCR assay was used for the study of bacterial load. The statistical analysis is sound

Response: Thank you for your revision of the paper.

General comments

General comment 1. The inclusion criteria were patients with a first diagnosis of intestinal KPC-KP colonization during a hospital admission at the time of recruitment or within the three prior months. It seems different groups for the analysis. Do the authors perform a separate analysis in both groups?

Response: Thank you. For this study both types of patients were recruited in the same cohort. A previous study indicated that the percentage of persistent carbapenemase-producing *K. pneumoniae* carriers at 6, 12 and 24 months after discharge from an acute-care hospital was

55%, 30% and 20%, respectively (Feldman et al.; 2013 <https://doi.org/10.1111/1469-0691.12099>). Based on these figures, when our study was designed back in 2016, we decided to also account for patients with a recent diagnosis of KPC-KP rectal colonization (3 prior months) to monitor both persistence of colonization and bacterial load in these patients. Certainly, as the reviewer suggests, both groups may represent different types of patients for the purpose of our study, due to relevant differences in clinical factors at the time of recruitment, including bacterial load of KPC-KP in intestinal carriers. Moreover, we have recently published a paper in *Microbiology Spectrum* derived from a different cohort study in our hospital, which confirms that there is an association between the timing of first detection of colonization and the risk of developing *Klebsiella pneumoniae* carbapenemase-producing *K. pneumoniae* infection in hospitalized patients (Cano et al.; Microbiology Spectrum, 2022, <https://journals.asm.org/doi/10.1128/spectrum.01970-21>), which further supports the idea that outpatients with a recent hospital admission and hospitalized patients at recruitment may represent different groups for the purpose of our study. Consistently, we show that there is a statistically significant difference in the median KPC-KP load between both groups (Supplementary Figure S4). For all these reasons, as the reviewer suggests, we performed and reported our multivariable analyses both in the global cohort of 80 patients and in the subcohort of 68 patients hospitalized at recruitment. This was done both for the competing risk Fine-Gray regression model for first KPC-KP-related infection episode versus death (**Table 2**) and for the multivariable Cox regression models of factors associated with all cause 90-day (**Table 3A, C**) and 30-day mortality (**Table 3B, B**). These analyses could not be performed in the subgroup of non-hospitalized patients due to the small sample size (N=12).

General comment 2. How do the authors control the sampling (rectal swabs and not perirectal swabs?) Include type and/or trademark of rectal swabs.

Response: The sampling was done by rectal swabs. Perirectal swabs were not allowed, which was specified in the study protocol. In our cohort study, this was also facilitated by the fact that all rectal swab sampling was performed by a single dedicated nurse. Regarding the trademark of the rectal swabs, it has been included in Methods, as follows:

Lines 175-179, marked copy:

Rectal swab samples were collected on day 0 of follow-up using eSwab rectal swabs (Copan, Brescia, Italy), which contain liquid Amies transport medium, and were submitted both to culture-based and qPCR-based quantification of the relative intestinal load of KPC-KP, based on the procedures originally published by Lerner et al. (27) (summarized in **Supplementary Figure S1**).

General comment 3: This group has experience with intestinal decontamination (reference 34). Was the use of this procedure considered an exclusion criteria?

Response: Colonized patients who received a clinical indication of selective intestinal decolonization with oral non-adsorbable antibiotics were included in the KLEBCOM cohort but were excluded from the present analysis, and this subcohort is described elsewhere: Pérez-Nadales et al.; Journal of Global Antimicrobial Resistance, 2022 (reference 23, <https://doi.org/10.1016/j.jgar.2022.04.010>). This information has now been included in Methods “Study design”, lines 142-144, marked copy)

General comment 4. The total number of patients is very low and this might have bias the results. Nevertheless, they did a rigorous analysis of the data obtained.

Response: We thank the reviewer for this appreciation. Indeed, the total number of patients is low, so we made an effort to control for confounding with a rigorous analysis. In addition, the rate of acceptance for participation and signing of the informed consent during the study period was high, so we consider the study has the added value of representing a real clinical context and experience with these patients in a tertiary hospital with endemicity by KPC-KP.

Specific comments

Specific comment 1: Title. I will recommend to use the acronym KPC instead "Klebsiella pneumoniae carbapenemase" This will make the title more readable.

Response: Thank you, we agree and have changed the title accordingly.

Specific comment 2: Line 54. Is this a specific 16S rRNA for K. pneumoniae or total 16S rRNA?

Response: We used primers specific for the *bla*_{KPC} gene and general primers for amplification of total 16S rRNA, as a means to estimate the number of total bacteria present in a patient's rectal swab. We have specified this information in Methods (section "Variables and definitions"), as follows:

Lines 159-164, marked copy The relative intestinal load of KPC-KP within the gut microbiota (RL_{KPC}) was defined as the percentage of the number of copies of the *bla*_{KPC} gene (representing KPC-KP) relative to 16S rRNA genes (representing total bacteria) estimated in total genomic DNA extracted from a patient's rectal swab obtained on day 0 of follow-up, according to qPCR analysis (see below).

Specific comment 3: Line 84. Immunological conditions? This might be out of the scope

Response: We agree with the reviewer, it is hypothetical. This has been removed from the abstract.

Specific comment 4: Line 193. Range of patient's age is very high. Were younger patients excluded?

Response: Yes, only adult patients were included. This has now been specified in Methods ("Study design" section, Line 126. Marked copy).

Specific comment 5: Line 223. Antibiotic susceptibility instead antibiotic sensitivity

Response: Corrected, thank you.

Specific comment 7: Line 227. There is no EUCAST criteria for tigecycline and K. pneumoniae. Do the authors used the PK-PD breakpoint for this antibiotic?

Response: The 2019 EUCAST breakpoints document (used in this study) includes explicit clinical categories of tigecycline for *Escherichia coli* and *Citrobacter koseri*, but in note 3/A also indicates that "for other Enterobacterales, the activity of tigecycline varies from insufficient in *Proteus* spp., *Morganella morganii* and *Providencia* spp. to variable in other species. For more information, see http://www.eucast.org/guidance_documents/". In this website, the

“Guidance Document on Tigecycline Dosing” (from 2018) it is indicated that Monte Carlo simulation analyses with isolates of carbapenemase-producing *Klebsiella pneumoniae* found that a cumulative fractional response rate of more than 90% was achievable using higher-exposure dosing regimens with tigecycline (100 mg 12-hourly with or with a 200 mg loading dose). PK-PD of tigecycline at these high doses predicts that MDR-strains (including specifically *K. pneumoniae*) with tigecycline MICs of up to 1 mg/l will respond to treatment. Accordingly, this PK-PD breakpoint was considered for clinical categorization of the isolates evaluated in this study. Importantly, this information has not changed in the 2022 EUCAST Tables.

Specific comment 8: Line 236. Was the antibiotic treatment influence this figure?

Response: Lines 235-239 in our article stated: "KPC-KP all-site infections at day 30 were observed in 32 (40.0%) patients, meaning that the time elapsed from rectal swab collection on day 0 of follow-up to infection onset was below 30 days in all except one patient (who developed infection on day 89), with a median (range) time of 4 (2-7) days."

Based on the previous comment (**specific comment 7**) made by the reviewer, we are wondering whether this question may refer to the treatment with Tigecycline and whether the reviewer is concerned that the lack of a breakpoint point for tigecycline may have influenced the evolution of infections. If this is the case, we have provided an answer in the previous comment. Otherwise, we kindly ask the reviewer to clarify this question.

Specific comment 9: Line 363-365. This can be due to the progressive elimination of Firmicutes and increase of Proteobacteria with aging

Response: Very interesting observation, thank you. We have re-written this paragraph of the Discussion in response to Reviewer 1 (major comment 4) and have added a reference to this idea also for a more complete discussion. As follows:

Lines 411-428, marked copy: Age-related alterations in the gut microbiome are influenced by factors such as progressive physiological deterioration, and lifestyle-linked factors including diet, medication and reduced social contact (38). In people over the age of 70, a study reported a decrease in anaerobic bacteria such as *Bifidobacterium* spp., which has a role in the stimulation of the immune system and metabolic processes, and an increase in *Clostridium* and Proteobacteria (39). Hospitalization may exacerbate microbiota dysbiosis in frail, older people as a result of medication and exposure to healthcare invasive procedures. The niches left open within the intestinal microbiota may be occupied by overgrowth of microorganisms that have pathogenic potential at high densities, with the healthcare personnel acting as a transmission vector in the hospital environment. Members of the Proteobacteria, such as *Escherichia coli* or *K. pneumoniae*, normally represent less than 2% of the microbiota, but in a dysbiotic state these bacteria can represent up to 30% of the total species (18). We hypothesize that an increased intestinal load of KPC-KP within the gut microbiota in our clinical context may behave as a surrogate marker of debilitated health condition because of frailty, higher burden of comorbidities, or age-related alterations in the gut microbiome and immune dysfunction (immune senescence), which may in turn lead to higher risk of infection or death.

Spectrum02728-21R1_Response to Reviewers

Specific comment 10: Table S3. Include S / I / R in this order.

Response: Corrected, thank you.

Staff Comments:

Yours faithfully,

Elena Pérez-Nadales, PhD (corresponding author)

E-mail address: elena.pereznadales@imibic.org.

Telephone: (+34) 957 213819

May 20, 2022

Dr. Elena Pérez-Nadales
Maimonides Biomedical Research Institute of Cordoba, Reina Sofía University Hospital, University of Cordoba
(IMIBIC/HURS/UCO)
Córdoba
Spain

Re: Spectrum02728-21R1 (Prognostic significance of the relative load of KPC-producing *Klebsiella pneumoniae* within the intestinal microbiota in a prospective cohort of colonized patients.)

Dear Dr. Elena Pérez-Nadales:

Your manuscript has been accepted, and I am forwarding it to the ASM Journals Department for publication. You will be notified when your proofs are ready to be viewed.

Sincerely,

Ahmed Babiker
Editor, Microbiology Spectrum
